# Habits and Persistent Food Restriction in Patients with Anorexia Nervosa: A Scoping Review

**DOI:** 10.3390/bs13110883

**Published:** 2023-10-25

**Authors:** Ismara Santos Rocha Conceição, David Garcia-Burgos, Patrícia Fortes Cavalcanti de Macêdo, Carina Marcia Magalhães Nepomuceno, Emile Miranda Pereira, Carla de Magalhães Cunha, Camila Duarte Ferreira Ribeiro, Mônica Leila Portela de Santana

**Affiliations:** 1Graduate Program in Food, Nutrition and Health, School of Nutrition, Federal University of Bahia, Salvador 40110-907, Brazil; ismararocha@ufba.br (I.S.R.C.); macedopatriciafortes@gmail.com (P.F.C.d.M.); 2Department of Psychobiology, The “Federico Olóriz” Institute of Neurosciences, Biomedical Research Centre, University of Granada, 18071 Granada, Spain; davidgb@ugr.es; 3Department of Psychology, Metropolitan Union of Education and Culture, Lauro de Freitas 42702-420, Brazil; cari.magalhaes@gmail.com; 4Clínica Elpis Addictions Treatment Service, Salvador 41940-650, Brazil; emilenut@yahoo.com.br; 5School of Nutrition, Federal University of Bahia, Salvador 40110-907, Brazil; carlamagalhaesc@gmail.com (C.d.M.C.); camiladuartef@ufba.br (C.D.F.R.); 6Graduate Program in Food Science, Faculty of Pharmacy, Federal University of Bahia, Salvador 40170-115, Brazil

**Keywords:** eating disorders, cognitive neuroscience, habit, anorexia nervosa, food restriction, scoping review

## Abstract

The aetiology of anorexia nervosa (AN) presents a puzzle for researchers. Recent research has sought to understand the behavioural and neural mechanisms of these patients’ persistent choice of calorie restriction. This scoping review aims to map the literature on the contribution of habit-based learning to food restriction in AN. PRISMA-ScR guidelines were adopted. The search strategy was applied to seven databases and to grey literature. A total of 35 studies were included in this review. The results indicate that the habit-based learning model has gained substantial attention in current research, employing neuroimaging methods, scales, and behavioural techniques. Food choices were strongly associated with dorsal striatum activity, and habitual food restriction based on the self-report restriction index was associated with clinical impairment in people chronically ill with restricting AN. High-frequency repetitive transcranial magnetic stimulation (HF-rTMS) and Regulating Emotions and Changing Habits (REaCH) have emerged as potential treatments. Future research should employ longitudinal studies to investigate the time required for habit-based learning and analyse how developmental status, such as adolescence, influences the role of habits in the progression and severity of diet-related illnesses. Ultimately, seeking effective strategies to modify persistent dietary restrictions controlled by habits remains essential.

## 1. Introduction

Anorexia nervosa (AN) is a severe, multifactorial eating disorder characterised by the restriction of caloric intake leading to significantly low body weight, intense fear of gaining weight, and disturbance in how body weight or shape is experienced [1]. Notable elements of AN are the absence of recognition of the severity of low body weight and the persistence of behaviour that interferes with weight gain, such as food restriction, excessive physical activity, and purging [2].

Estimates of lifetime AN prevalence are 4% for women and 0.3% for men. Although the overall incidence rate of AN has been stable in recent decades, its incidence among people aged <15 years has increased [3]. AN tends to present a persistent course, with a median duration of 10 years [4] and a standardised mortality rate of 5.9 (95% CI 4.2–8.3), a risk almost six times higher than expected in the population of origin [5]. Higher mortality rates of 15.9 (95% CI 11.6–21.4) have been observed in hospitalised patients with AN with complications from severe malnutrition [6].

Despite the recognition and awareness of AN, the progress achieved so far has not been adequate to deal with its severity [7]. Moreover, despite extensive scientific research, there is still a lack of consensus regarding the underlying aetiology of AN. An important distinction in exploring the mechanisms lies between individuals in the early stages of the illness and those with a chronic course. In this sense, reinforcement and habit formation mechanisms have been identified as significant factors contributing to the transition from initial manifestations to the long-term persistence of the disease. Indeed, regarding enduring and maladaptive food restriction in AN, habit formation is believed to play a crucial role in the chronicity and severity of AN [8,9,10,11,12,13,14,15,16,17,18,19]. However, the potential role of habits in the symptomatology of AN remains to be empirically confirmed.

Habits are learned behaviour patterns that become fixed and require minimal conscious supervision. As a result, cognitive resources are freed up for other activities [20]. These behaviours, initially directed toward a specific goal, gradually become automatic, less sensitive to outcome, and more closely related to the surrounding circumstances. Thus, with sufficient repetition, habits will be elicited by a cue. From a learning standpoint, habits have the same structure as reflexes, involving a stimulus–response association. However, while reflexes are innate, habits must be learned (see [21]). By recognising the potential of habit processes as a novel research avenue for understanding the mechanisms of AN, this scoping review aims to map the literature on the contribution of habits to food restriction in AN. The research questions (RQs) for this scoping review are provided below and described in the protocol:(RQ1) What evidence is available on how habits contribute to food restriction in AN?(RQ2) What methods are used to evaluate habits related to food restriction in AN?(RQ3) What are the mechanisms of habit learning in food restriction in AN?

## 2. Materials and Methods

### 2.1. Protocol and Registration

This scoping review followed the PRISMA Extension for Scoping Reviews (PRISMA-ScR) [22]. The PRISMA-ScR checklist used can be found in Appendix A. A scoping review protocol was developed based on the guidelines of the Preferred Reporting Items for Systematic Reviews and Meta-Analysis Protocols (PRISMA-P) [23] and the Joanna Briggs Institute manual [24]. The protocol was registered in the Open Science Framework on 8 January 2022 (https://osf.io/d2jns).

### 2.2. Eligibility Criteria

The review adopted the Population, Concept, and Context (PCC) framework, focusing on the question and the following eligibility criteria: (P) people with anorexia nervosa, (C) eating habits and food restriction, and (C) location/environment of participant recruitment, including country and demographic characteristics. The specific inclusion and exclusion criteria for study selection are detailed in Table 1. This review considered various studies, including clinical trials, observational and qualitative studies, experimental design for animal and human studies, reviews, opinions, case reports/series, and conference abstracts. In vitro studies, protocols, and books were excluded. The search for records was conducted without restrictions on language or publication year.

### 2.3. Sources of Information and Search Strategy

The search strategy was developed after consulting a health librarian and applied to the following databases, accessed on 19 August 2022: MEDLINE via PubMed, EMBASE via Elsevier, Latin American and Caribbean Health Sciences Literature (LILACS), Cochrane Library Databases, PsyARTICLES, Web of Science, and the Cumulative Index to Nursing and Allied Health Literature (CINAHL). Additional sources of evidence were searched in grey literature (Google Scholar) and manually in the reference lists of the studies. According to the recommendations of Greenhalgh and Peacock [25], five specialists were identified and consulted to track published studies not found in the searched databases.

A limited search was initially performed in Medline/PubMed to identify potentially relevant documents. At this stage, the terms “habit”, “anorexia nervosa”, and “food restriction” were identified in the titles, abstracts, and keywords of the recovered documents. Finally, all terms and their synonyms were selected from Medical Subject Headings (MESH), Embase Subject Headings (Emtree), Health Sciences Descriptors (DeSC), and the APA Thesaurus of Psychological Index Terms and included: “Anorexia Nervosa” OR “Anorexia Nervosas” OR “Atypical Anorexia nervosa” AND “Habits” OR “Habit*” OR “Habit Disturbances” OR “Habit Disturbance” OR “Learned habit” OR “Conditioned habit” OR “Habitual behavior” OR “Habit systems” OR “Habit strength” OR “Habit formation” OR “eating habits” OR “Food restriction” OR “Food refusal” OR “Dietary restraint” OR “Dieting”. They were then added to the search strategy to be applied in all databases using the Boolean operators “AND” and “OR”. The strategy can be found in the Appendix A. The online software EndNote classic version https://endnote.com/ (accessed on 19 August 2022), a reference manager, was used to remove duplicates.

### 2.4. Selection of Sources of Evidence

Before screening titles and abstracts, the Rayyan online tool organised the electronic database search (https://www.rayyan.ai/, accessed on 5 September 2023). Then, a pilot test was carried out on 25 randomly selected records. When 80% of agreement was reached, two independent reviewers began the selection of all retained records by title and abstract. Finally, the reviewers examined the records by performing a full reading and, based on the eligibility criteria, selected the studies included in this review. Disagreements in the two stages of selecting evidence sources were discussed and resolved with two reviewers.

### 2.5. Data Extraction and Items

Data extraction was performed using a spreadsheet elaborated in Microsoft 365 Excel version 2208, developed and tested in an independent pilot extraction of a random sample of three selected studies performed by two team members. The process of filling out the form was frequently reviewed in discussions with the project team. Next, the data were extracted by the two reviewers who had independently extracted the relevant data. Two reviewers then verified the quality of the data for consistency.

The information extracted consisted of the characteristics of the study (author, year of publication, country), the objective of the study, sample characteristics (sample size, age, sex, race, and socioeconomic status of the participants, duration of the disease), place of recruitment, method of diagnosis of the eating disorder, evaluation of the symptomatology of severity of the disease or method of evaluation of food restriction, habit evaluation method, and main results. The corresponding authors were contacted for missing information by e-mail, and responses were received for six sources of evidence.

### 2.6. Summary of Results

The findings were presented using a narrative format, tables, and figures. The characteristics of the sources of evidence were reported to assess the available evidence on how habits contribute to food restriction in individuals with AN. The characteristics also included information on the countries and publication years of the studies, the populations studied, and the method of analysing food restriction and/or severity in AN. To examine how habits were measured, we categorised the experimental methods used to operationalise the habit related to food restriction in individuals with AN. Based on substantial research, we investigated which mechanisms of habit-learning contribute to food restriction in AN, addressing behavioural and neural aspects in this context.

## 3. Results

### 3.1. Selection of Sources of Evidence

The search strategy across seven databases recovered 6137 records. After removing duplicates, 4427 citations were assessed based on their titles and abstracts. Of these, 57 publications were selected for a full reading. After applying the eligibility criteria, 22 publications were excluded, leaving 35 for the next stage. In addition, following an evaluation of grey literature, a manual search in the reference lists of the retained studies, and a consultation with five specialists, six publications were added to the thirty-five included in the previous stage. Thus, the final search included 35 studies, comprising a total of 41 publications [9,10,11,12,13,14,15,16,17,18,19,26,27,28,29,30,31,32,33,34,35,36,37,38,39,40,41,42,43,44,45,46,47,48]. To account for the total number of publications, two studies contributed both an article and a conference abstract [30,32,49,50]. The other two studies also comprised three reports each [31,51,52,53,54]. The selection process and reasons for exclusion are detailed in an adapted PRISMA flowchart (Figure 1).

### 3.2. Characteristics of Evidence Sources

The description of the main characteristics of the 35 studies included in this review is summarised in Appendix A. Among them, 20 were empirical studies (randomised clinical trials [RCTs] and non-randomised clinical trials [NRCTs], experimental, cross-sectional, and longitudinal studies) and 15 were reviews. Regarding the distribution of studies around the world, the highest number was identified in the USA (n = 19), followed by the United Kingdom (n = 6), Germany (n = 5), France (n = 2), New Zealand, and Canada (n = 1 in each country). One study (Godier et al. [15], study 1 and study 2) was developed in the United States and the United Kingdom. According to the broader bibliometric research trends, the first study, published in 2006, was a narrative review [19], followed by a substantial chronological increase in publications, especially from 2018 to 2021 (Figure 2).

### 3.3. Characteristics of the Participants

Of the total observational studies, one did not include a healthy control group [40]; one experimental study [32] conducted research on humans and on an animal model of AN; three clinical trials compared an intervention versus sham/standard therapy in patients with AN (Appendix A). The sample sizes of the AN groups ranged from 10 [28] to 80 [29] individuals, whose time course of the disorder was from 0.7 to 15.7 years. Samples ranging from 12 [44] to 54 [35] individuals comprised the control groups. Most studies (n = 16) included exclusively female individuals [9,15,26,27,28,29,31,33,34,35,37,38,39,42,43,44] and adolescent and adult participants (n = 9) [27,29,31,34,37,38,40,42,43]. The white race was predominant in studies that evaluated this demographic variable (Appendix A).

Participants with AN were recruited in hospitals and or outpatient services specialised in the treatment of eating disorders (n = 17) [9,26,27,28,29,30,31,32,33,34,35,37,39,40,42,43,44] and in other different catchment spaces [15,38] (Appendix A). Diagnostic confirmation was performed using different tools, with a predominance of the Eating Disorder Assessment for DSM-5 (EDA-5) [9,28,31,38,39] (n = 5) followed by the Eating Disorder Examination (EDE) (n = 4) [15,26,30,43] and the Structured Interview for Anorexia and Bulimia Disorders (SIAB-EX) (n= 3) [27,37,42] (Appendix A). To assess the symptomatology of AN’s severity, most studies adopted the Eating Disorder Examination Questionnaire (EDE-Q) (n = 10) [9,15,28,31,33,34,35,38,39,43], followed by the Eating Disorders Inventory-2 (EDI-2) (n = 4) [27,32,37,42], the Yale–Brown–Cornell Eating Disorder Scale (YBC-EDS) (n = 3) [26,30,31], and the Three-Factor Eating Questionnaire (TFEQ) (n = 3) [43,44,51].

### 3.4. Results of Sources of Evidence

#### 3.4.1. Methods for Evaluating Habits

The evaluation of habits related to food restriction in individuals with AN was based on distinct methods (Figure 3), namely: (a) scales (n = 4) [9,27,39,40], (b) neuroimaging (n = 11) [26,28,29,30,31,33,35,38,42,43,44], (c) experimental and behavioural tasks (n = 3) [15,32,34], and (d) neuroimaging and experimental and behavioural tasks (n = 2) [36,37].

##### Scale

Four empirical studies used the Self-Report Habit Index (SRHI) scale to operationalise habit strength. Coniglio et al. [40] measured the habit strength of a single behaviour (food restriction), while Steinglass et al. [39] and Davis et al. [9] measured behaviours composed of four categories relevant to the eating disorder: (1) restrictive eating, (2) compensatory behaviours, (3) delay in eating, and (4) rituals around eating. In addition, these two studies explored the effect of automaticity using an SRHI subscale, The Self-Report Behavioural Automaticity Index. A study by Seidel et al. [27], in addition to using 10 of the 12 items of the SRHI scale, also evaluated habit frequency through ecological momentary assessment for the categories of specific to eating disorder (food intake) and non-specific to eating disorder (hygiene).

##### Neuroimaging

In most of the empirical studies included in the review, functional magnetic resonance imaging (fMRI) was the neuroimaging technique adopted to assess habits [28,29,31,33,37,38,42,43,44]. In contrast, three studies used structural magnetic resonance imaging [28,33,35] and two used diffusion magnetic resonance imaging (dMRI) [26,30].

Of the studies that employed fMRI, two [36,37] also adopted tasks (two-step decision, instrumental motivation task), performed following the fRMI, to assess participants’ habitual behaviours (see details in the sub-item “experimental and behavioural tasks”). Other tools associated with fMRI were used to explore brain activity in habit-related areas via food cues [44], self-control/decision-making [42], and food choice tasks [28,29,31,33,36,43]. For the most part, areas related to reward circuits (nucleus accumbens—Nacc, orbitofrontal cortex—OFC) and habit circuits (dorsolateral prefrontal cortex—DLPFC, dorsal anterior cingulate cortex—dACC, caudate, putamen, and dorsal striatum—DS) were investigated in the studies included in this review. The two studies [26,30] that investigated habits by dMRI also used tractography to analyse the region of interest (ROI), obtaining microstructure maps of the habit decision-making circuit in the sensory–motor area (SMA) to the putamen pathway. Also, to discern the microstructure of white matter, Murray et al. [26] used a neurite density orientation and imaging scattering tool (NODDI) in addition to the NODDI analytical model of absolute tissue density (ABTIN).

##### Experimental and Behavioural Tasks

In three studies [15,32,34], habits were assessed through tasks/paradigms. Godier et al. [15] and Favier et al. [32] used a slip-of-action neurocognitive test, an abbreviated version of the original “Fabulous Fruit Task”, and validated it. The test consists of three phases: an instrumental learning stage, a simple outcome devaluation choice test, and an initial test versus “action slip” test (go/no-go control task). The use of fruit images as a stimulus for the test was adopted by Godier et al. [15] (study 1). Also, non-food stimuli such as animal imaging were adopted in the execution of the test [15,32] (study 2). Moreover, Godier et al. [15] (study 2) evaluated the avoidance of habits in AN using a noise avoidance task, an adapted version of the shock avoidance task. The task consists of four stages: a brief training session, a devaluation extinction test (devaluation sensitivity test), an extended training session, and a final extinction evaluation (habit test).

One study [34] adapted a Pavlovian-instrumental transfer paradigm (PIT) to investigate Pavlovian learning and the impact of stimuli conditioned to high and low-calorie food rewards in patients with AN. The PIT employed in the study comprised three experimental phases with food-related stimuli to examine how conditioned stimuli can activate instrumental responses.

A two-step decision task was administered in a study by Steinglass et al. [36]. Each participant completed a decision task involving playing monetary or food outcomes. This task evaluated model-free and model-based learning to assess habitual and goal-directed behaviours, respectively. An instrumental motivation task is based on the evaluation of stimuli that predict manual reward, and it was employed in a study by Steding et al. [37]. It was divided into an anticipation phase, a motor response phase, and a feedback phase. Regardless of reward level, high effort suggested a habitual response in the test.

### 3.5. Habit Mechanism in AN Food Restriction

The studies included in this review agree that food restriction is a central feature of AN. This restriction is developed and maintained by an interplay of behavioural and neural mechanisms related to habit learning and formation. In AN, positive social feedback can initially reinforce behaviour such as food restriction. Over time, this behaviour becomes an ingrained habit, independent of the initial rewards, and can be maintained even when the consequences become negative, such as damage to physical and emotional health [18,27].

#### 3.5.1. Behavioural Mechanisms

Highly structured and inflexible habits can be explained by behavioural mechanisms that include reinforcement learning, as reported in seven studies [9,15,27,32,34,39,40]. These involve rituals and rigid eating routines around food [9,27,39,40], decision-making, including computational tasks [15,32], and conditioned stimuli [34].

Walsh [18] proposes that food restriction emerges as a habit that persists due to reinforcement and conditioning. Conditioned reinforcement may be related to habit development in AN, even when the consequences are devalued.

Favier et al. [32] investigated habit formation in two groups of ED patients (AN-R and binge eating/purging eating disorder patients) using a computer-based neurocognitive task. The results showed that individuals with AN-R persistently responded to stimuli associated with devalued outcomes during the “action slip” stage. Furthermore, the study found that the percentage of responses to undervalued outcomes was correlated with a lack of cognitive flexibility in individuals with AN. However, the study conducted by Godier et al. [15] did not identify significant differences (*p* > 0.05) between the groups. After the devaluation outcomes, all groups were equally capable of retaining inadequate answers. This research emerged from two samples that recruited participants from the United States and the United Kingdom. In sample 1, participants with AN (restrictive and binge/purge) and HCs completed an action slip paradigm designed to assess habits based on reward. Sample 2 contained participants with restrictive AN, in recovery from restrictive AN, and HC participants who completed the action slip and avoidance paradigms designed to assess aversive habits.

Another study investigated conditioned stimuli, and the results showed that participants with AN and HCs responded similarly to high- and low-calorie foods during tests. However, a stronger association between instrumental behaviour and low-calorie foods was identified in individuals with AN. Furthermore, it was observed that participants with AN were less aware of Pavlovian contingencies [34].

Four studies [9,27,39,40] were conducted to analyse habit strength in the context of rituals and rigid eating routines involving food. In these studies, we sought to understand how behavioural automation influences people’s structured and inflexible eating patterns around food. Coniglio et al. [40] evaluated habit strength in groups of patients with AN and atypical AN. The results showed that habit strength predicts dietary restriction in patients with AN, explaining 27.9% of the variance in the Eating Pathology Symptoms Inventory. Furthermore, the results indicated that habitual dietary restriction was associated with clinical impairment in individuals with AN. On the other hand, Seidel et al. [27] found no increase in habit strength in acutely underweight patients with AN compared to HCs.

Analysing the four domains (restrictive eating, compensatory behaviour, rituals around eating and delay in eating) of the SRHI scale, Davis et al. [9] reported that individuals with AN (restrictive and binge eating/purging subtypes) presented greater habit strength (*p* < 0.05) compared to HCs. These results were evidenced in the total SRHI score and each evaluated domain. In a randomised controlled clinical trial performed by Steinglass et al. [39], the researchers investigated the effectiveness of a behavioural intervention, Regulating Emotions and Changing Habits (REaCH), compared to supportive psychotherapy, measuring the reduction in habit strength through the SRHI. The results showed significant changes in the SRHI domains of participants with AN, except in those about rituals around food (*p* > 0.05).

Seidel et al. [27] and Davis et al. [9] found that habitual behaviours can contribute to the persistence and severity of AN symptoms, and habit strength can increase as a function of the duration of the disease [9]. In terms of secondary outcomes, REaCH treatment had a significant (*p* < 0.05) effect on AN severity, as assessed through the EDE-Q global score [39]. Three studies [9,27,39] addressed food intake related to habit strength. According to Seidel et al. [27], the chance of reporting habits related to food intake was 1.78 times greater in patients with AN compared to HCs. REaCH significantly impacted (*p* < 0.05) the strength of habits individuals reported after treatment. Therefore, a higher degree of habit strength, as measured by the SRHI, was associated with reduced food intake during the laboratory meal [39]. However, Davis et al. [9] reported no significant association (*p* > 0.05) between habit strength and caloric intake.

For a more comprehensive understanding of AN, it is essential to adopt a general approach that considers this disorder’s behavioural and neural aspects. Grasping the interplay between these two elements can facilitate the development of more effective treatment strategies.

#### 3.5.2. Neural Mechanisms

Neural mechanisms include changes in activity and connectivity between the brain regions involved in reward [10,11,12,13,14,16,17,19,28,29,33,43,51] and psychological and emotional regulation [10,11,12,17,19,20,45,48]. Regarding the instrumental learning of inappropriate eating behaviour that drives weight loss in AN, some studies show that initially, these behaviours are aimed at goals (action–outcome learning) and are processed in the reward system, which includes the ventral striatum (VS), Nacc, and ventromedial prefrontal cortex (VMPFC) cortical regions [13,18,41,42]. Over time, restrictive eating in AN is repeatedly and persistently engaged in, leading to a shift in focus from goals to habitual behaviours (stimulus–response learning), which, when reinforced, become increasingly fixed and automated, as reported in 10 studies [16,17,18,19,20,41,45,46,47,48]. In addition, six studies [11,12,17,19,20,46] showed that the more automated a behaviour is in response to a specific stimulus, the less conscious the decisions made by the individual with AN will be, resulting in dysfunctional and cognitively inflexible food choices.

Nine review studies showed that persistent food restriction could lead to changes in functional connectivity in the areas of the brain that regulate habits. These studies focused on investigating the white matter [46,47], the dorsolateral prefrontal cortex (DLPFC), and the basal ganglia and their structure called the dorsal striatum (DS) [10,13,14,16,18,19,20] (see Figure 4).

For example, a study by Foerde et al. [43] analysed the DS and DLPFC using fMRI and found that low-fat food choices resulted in greater DS activation (*p* < 0.05) and that activity in frontostriatal circuits correlated with actual food intake (the food in a meal eaten the next day) (*p* < 0.05) in the AN group compared to HCs. Steinglass et al. [29] found no significant difference in the involvement of the ventral or dorsal striatum in food choice between groups of adolescents with AN and HCs.

Using high-frequency repetitive transcranial magnetic stimulation (HF-rTMS) in the DLPFC, Muratore et al. [28] found a reduction in fat avoidance among patients with AN compared with those undergoing sham control treatment (*p* < 0.05). On the other hand, in a randomised clinical trial by Dalton et al. [33], the use of HF-rTMS in the DLPFC did not influence fat-related food choices. However, there was a decrease in self-controlled choices and an increase in tasty and unhealthy food selection (*p* < 0.05) in the group that received real HF-rTMS compared to the sham. Furthermore, in Muratore et al. [28], when applying the psychophysiological interaction model (PPI), a more robust connectivity was observed for low-fat foods (*p* < 0.05), with a peak in the DLPFC.

Investigating anomalies in the shape and volume of the basal ganglia in individuals with AN and HCs, the results of a study by Leppanen et al. [35] showed significant differences in the shape of the left caudate and bilateral globus pallidus between groups but no differences were identified in the volume of the basal ganglia. When evaluating the white matter, the results of a study by Murray et al. [26] observed no significant differences in bilateral neurite density, bilateral neurite orientation, and bilateral myelin density (all associations *p* > 0.05) in white matter tracts between the sensory–motor area (SMA) and the putamen, suggesting that in weight-restored patients with AN, it is not possible to identify a functional abnormality in the white matter of the habit circuit. However, Tadayonnejad et al. [30], in turn, provided evidence that there is an increase in the volume of bilateral white matter tracts in the habitual decision-making circuit, associated with the severity of symptoms of ritualistic behaviours in adolescents and adults with AN (both *p* < 0.05).

Considering decision-making in the habit circuit, Steding et al. [37] identified two subgroups of patients with AN, with the subgroup of patients oriented by habit (hAN) being characterised by a greater use of the decision-making model-free decision test (*p* < 0.05), using an instrumental motivation task to assess habitual behaviour. Also, review studies by Frank, Shott, and DeGuzman [12] and Steinglass and Walsh [19] suggested dysfunctions in the neural circuits involved in habit-learning and decision-making, especially in the DS and DLPFC regions. This finding was confirmed by a study by Foerde et al. [43], which revealed that the dorsal striatum is responsible for guiding food choice decisions in patients with AN but not in HCs.

According to nine review studies [11,13,14,16,18,20,45,46,47], internal and external factors, such as the relief of adverse effects, the pleasure associated with the feeling of hunger and the act of eating restricted food, and the associated positive social reinforcement of food control, weight loss, and changes in physical appearance, can make food restriction rewarding. On the other hand, some studies agree that in the processing of stimulus–response learning, hunger, stress, and anxiety associated with food control can lead to neurotransmitter-related [11,16,41] and hormonal [11,16,17,45] changes, stimulating and reinforcing weight loss behaviour.

The role of hunger in habit formation in AN was discussed in five review studies [11,41,45,47,48]. These studies suggested that individuals with AN may be better able to maintain a restrictive diet long enough to form habitual behaviour, as increased serotonin or 5-hydroxytryptamine (5-HT) activity may reduce hunger and increase satiety and anxiety [41]. In addition, Guarda et al. [45] conducted a study revealing that hunger can affect the functioning of the hypothalamic–pituitary–adrenal (HPA) and hypothalamic–pituitary–gonadotropic (HPG) axes [45], which play a role in stress control. Regarding hunger, Park, Godier, and Cowdrey [47] proposed in their study that severe restriction of food intake can become compulsive and persist despite negative consequences in both the short and long term. Furthermore, Rufin and Steinglass [11], Lloyd et al. [41], O’Hara, Campbell, and Schmidt [16], and Marsh et al. [48] suggested that ritualistic and compulsive behaviours can persist through habitual control, regardless of their outcome. Lloyd et al. [41] highlighted that compulsive behaviour in individuals with AN may arise from a disruption in the goal-directed circuit, resulting in the deactivation of the frontostriatal pathway. This circuit would typically counteract the influence of the habit circuit when the behaviour is inappropriate. For instance, Tadayonnejad et al. [30] found a positive correlation between the severity of ritualistic/compulsive symptoms and the total number and total volume of bilateral tracts when combining adolescent and adult (both *p* < 0.05) AN groups. Moreover, a review by Godier and Park [46] proposed a mechanism involving habitual and compulsive behaviours contributing to and maintaining weight loss in AN.

Four review studies [11,16,17,45] discussed the relevance of stress in developing habits. These studies suggest that stress can induce changes in neurotransmitter systems, including dopamine (DA) and glutamate [11,17,45], resulting in alterations in dendritic morphology in DLPFC [45]. In addition, O’Hara, Campbell, and Schmidt [16] highlight the importance of DA in habit-learning as it receives and integrates excitatory signs that can lead to the formation of compulsive and ritualistic behaviours.

For Lloyd et al. [41], stress and anxiety are associated with an increased volume and activity of the dorsal striatum, that is, using the habit system to the detriment of the goal-directed system in individuals with AN. According to the researchers, high anxiety levels can reinforce the anxiolytic effects of food restriction, further reducing the function of the goal-directed system [41]. Rufin and Steinglass [11], in turn, highlight that emotional factors, such as anxiety, can influence repetitive and stereotyped eating behaviours (considered dysfunctional habits) and that individuals usually resort to these behaviours to deal with feelings of discomfort. Another point to be highlighted is the significant interaction between anxiety and the volume of white matter tracts in the habit circuit, showing a moderate positive correlation (*p* < 0.05), observed in the study by Tadayonnejad et al. [30]. It suggests that anxiety may influence habit development and maintenance by altering the integrity of white matter in the habit circuit.

Cholinergic interneurons are essential modulators of the striatal network and express both the vesicular acetylcholine transporter (VAChT) and the vesicular glutamate transporter type 3 (VGLUT3). Favier et al. [32] investigated the role of acetylcholine and glutamate release by cholinergic interneurons in habit formation and maladaptive eating in genetically engineered mice that no longer expressed VAChT (VAChTcKO mice) or VGLUT3 (VGLUT3cKO mice). The results showed that the inactivation of acetylcholine release by cholinergic interneurons induced a deficit in behavioural flexibility, increasing the predisposition of mice to habit formation.

## 4. Discussion

### 4.1. Summary of Evidence

#### 4.1.1. (RQ1) Evidence Available

A scoping review is a valuable tool to assess the breadth of existing literature, synthesise research findings, and identify knowledge gaps that must be addressed [56]. The present scoping review included 35 studies focusing on the emerging field of neurobiological research concerning habit-learning in AN, published from 2006 to 2023. Our findings provide important information on research trends, study designs, and participant characteristics, serving as a valuable resource for researchers looking to understand the role of habit in food restriction among individuals with AN.

The first study to discuss habits in AN was the review by Steinglass and Walsh in 2006 [19], which extended the neurocognitive model of obsessive–compulsive disorder to explain the mechanisms perpetuating AN. Unlike the review carried out in 2006, our review advances this knowledge by including studies in which the investigation of persistent restriction as a habit is observed in specific samples of people with AN. In these studies, neuroimaging is adopted to identify specific structural alterations of learning and of the habit loop for measuring brain and behavioural activity. In addition, new techniques that can help elucidate mechanisms and interventions in specific brain areas related to habits are also proposed.

Among the retained publications, the study by Walsh [18] stands out. It was published seven years after the first review on the subject and presented a habit-centred learning model based on stimulus–response learning, which helped to explain the persistence of AN. Also, this study represents a significant advancement in understanding biobehavioural mechanisms in individuals with AN. Foerde et al. [43] conducted the first empirical study to utilise fMRI associated with a task (food choice task) to examine the neural circuits involved in food decision-making. This study showed that people with AN have access to the dorsal striatum when deciding what to eat; this was a relevant finding for understanding the neural mechanisms related to this eating disorder.

Starting in 2018, there occurred a significant increase in research studies on habit-learning and food restriction in AN which employed innovative methodologies to investigate both neural [26,27,28,29,30,31,33,35,36,37,38] and behavioural [32,34,39] aspects. However, the behavioural mechanisms and brain circuits underlying habits in this eating disorder still need to be entirely understood.

Despite the lack of clarity on these mechanisms, evidence from this review suggests that understanding these processes has important implications for developing more effective treatment strategies [28,33,39]. Targeting the behavioural aspects and neural circuits involved in the creation of dysfunctional habits makes it possible to help individuals with AN establish a healthier relationship with food. Thus, different and promising approaches have been empirically tested to explain how neural and behavioural mechanisms are affected by treatment. To this end, randomised clinical trials included the HF-rTMS technique [28,33], which aims to modulate the DLPFC, and REaCH [39], which consists of a combination of behavioural techniques.

#### 4.1.2. (RQ2) Habit-Related Measures of Food Restriction in AN

This review included studies that used various methods to assess habits, such as scales [9,27,39,40], neuroimaging [26,28,29,30,31,33,35,38,42,43], and experimental and behavioural tasks [15,32,34,36,37,44], as well as a combination of these methods.

Research has consistently shown that habit strength is a reliable predictor of various health behaviours [57,58]. The SRHI tool, developed by Verplanken and Orbell [59], is a robust assessment to measure the strength of habits. This scale evaluates three primary aspects: the frequency of engaging in a behaviour (repetition), awareness and controllability (automaticity), and the extent to which individuals identify with the behaviour. The SRHI has been widely used to investigate the severity and persistence of different diseases. For example, studies have shown that habit strength related to restrictive food intake, as assessed by the SRHI, correlates with the severity of this restrictive behaviour in AN [9,40].

Neuroimaging was measured with fMRI, structural, and diffusion MRI, and investigated areas related to the habit circuit, such as the dorsal striatum, the dorsolateral prefrontal cortex, the basal ganglia, and white matter. fMRI has become a widely used method and is considered the gold standard for assessing neural function in individuals, regardless of whether they are healthy or sick [60,61,62]. This is due to its non-invasive nature, wide availability, and high spatial resolution. Studies mainly focused on individuals with AN and were conducted in conjunction with paradigms designed to identify specific brain regions, providing information about the neural mechanisms involved as well as determining functional connectivities between the different regions [63,64,65].

Indeed, these studies employ different tasks and paradigms to assess habits in patients with AN. Each provides crucial insights into the nature of habitual behaviours in this population. Behaviour can be studied in the laboratory through instrumental learning tasks involving outcome devaluation procedures and two-step decision tasks. These tasks have been associated with partially distinct neural substrates, indicating that different brain areas are involved in each type of learning [15,31]. The instrumental learning paradigm is widely used in scientific studies that aim to understand the cognitive and neural mechanisms involved in decision-making and habit-learning [66,67,68].

Relating to the use of the PIT paradigm in behavioural tasks with conditioned stimuli, few available studies measure the effects of these classically conditioned stimuli on behaviour. The experimental paradigm introduced by Corbit and Balleine [69] allows for a distinction between specific PIT and general PIT. It is an essential tool for studying how conditioned stimuli influence behaviour. Although there are still limited studies on the subject, this experimental approach provides relevant insights into understanding the interaction between classical and instrumental conditioning [70,71]. The use of PIT in humans is a relatively recent field of research and was adopted in the pilot study conducted by Vogel et al. [34], who showed the importance of investigating the effects of Pavlovian associations, providing a more enlightening view on behavioural and cognitive aspects in AN. In a recent review, PIT was characterised as an essential mechanism of action control that can characterise mental disorders, including food restriction in AN, in which stronger PIT effects elicited by low caloric stimuli were associated with increased disease severity [72].

Each method has its strengths as well as its limitations. For instance, fMRI studies can investigate the functioning of habit-related neural circuits. However, they cannot provide precise information about the habitual nature of behaviour since, in the habit-learning model, actions are more influenced by cues and less dependent on the outcome [39,73]. It is important to note that all tasks have limitations and should be interpreted cautiously. Therefore, combining these methods can offer a more comprehensive understanding of how habits may serve as a mechanism for disapproval of food choices in individuals with AN, considering the complex and multifaceted nature of this eating disorder.

#### 4.1.3. (RQ3) Mechanisms of Habit-Learning in AN Persistence

Habitual learning in individuals with AN involves complex and interactive mechanisms. Comprehending such mechanisms is crucial to understanding how habits are formed and persist and is fundamental in developing effective strategies for behaviour change. Below, we will present pertinent information about the behavioural and neural mechanisms involved in habitual learning.

#### 4.1.4. Behavioural Mechanisms Involving Habits

During the habit formation process, persistent restrictive eating behaviour in individuals with AN can be developed through reinforcement learning mechanisms [18,37,46]. This suggests that the association between stimuli and rewards is essential for developing reinforced and habitual behaviours. In this sense, food restriction becomes a rewarding behaviour for individuals with AN and contributes to the persistence and strengthening of their habits [15,20,31,32,36]. Furthermore, even when the initial rewards are no longer present, restrictive behaviours persist as a conditioned habit, performed without conscious effort [18,40]. In this condition, restrictive behaviour persists even when weight loss (initial goal) becomes less desirable. It means that specific AN-related behaviour, such as extreme dieting or excessive exercise, may have been initially reinforced by positive outcomes, such as receiving social praise for losing weight.

For instance, according to the reinforcement learning model, eating behaviour is a multifaceted interplay between homeostatic (physiological) and non-homeostatic (psychological and environmental) factors, encompassing events that form associations and become stored in memory. Consequently, sociocultural factors exert a substantial influence on food preferences and aversions. Notably, social pressure and various social contexts, such as family gatherings and peer dynamics, serve as influential models and provide numerous learning scenarios and stimuli. Thus, fatty food and becoming fat are often penalised socially. Additionally, sociocultural factors contribute to the establishment of body image ideals and beauty standards, imposing conformity to specific body types that can trigger disordered eating patterns and a skewed relationship with food. Another example is the availability and accessibility of food vary under the sway of sociocultural factors, including economic circumstances and geographical location, thereby influencing the range of dietary choices accessible to individuals.

However, behaviours become ingrained as habits over time, and the individual continues to engage in them regardless of external reinforcement. Initially, behaviours are learned and reinforced through positive outcomes, but as the habit cycle becomes established in the brain, behaviours can persist even without external rewards [18,27].

The reinforcement learning process recognises the role of the Pavlovian conditioned response, which is acquired from direct associative learning between a conditioned stimulus (e.g., a sound) and an unconditioned stimulus (e.g., food) [74]. Learning theories, such as Pavlovian (or classical) conditioning and instrumental (or operant) conditioning, are fundamental to understanding associations between mental representations of stimuli and responses in memory, which occur through internal processes, experiences, and interactions with the environment [8]. The pilot study conducted by Vogel et al. [34] provided important information on how individuals with AN and healthy controls respond to conditioned stimuli associated with low-calorie and high-calorie food rewards. The instrumental response, which aims to obtain a specific reward, was influenced by these conditioned stimuli in both groups. However, the most relevant finding was that in patients with AN, as the severity of eating disorder psychopathology increased, the instrumental response to low-calorie rewards also increased, suggesting that this disorder involves behaviour oriented to achieve a weight loss goal, which may be one of the factors that contributes to the persistent search for weight loss in AN. However, the authors point out that the findings may need to be confirmed in future studies with a more significant number of participants [34].

Habitual behaviours may result from conditioned associations between stimuli and responses. Studies have shown that the measure of habit strength positively correlates with the duration [9] and severity [9,40] of the symptoms observed in patients with AN. In addition, high frequency of habits in AN is highlighted as a potentially important factor in understanding the maintenance and expression of the disorder. This information improves the understanding of habits as a relevant component in the manifestation and maintenance of AN, which can provide valuable insights for developing more effective and targeted therapeutic interventions, such as REaCH [39].

By identifying habit strength as a significant factor in the persistence of restrictive behaviours, therapeutic interventions can be tailored to address this specific aspect in AN. They may include strategies to break automatic behaviour patterns and help patients develop self-control and flexibility skills around food. Eating rituals and routines can function as coping strategies or emotional self-regulation mechanisms to deal with stress, anxiety, or other negative emotions. In this sense, it is necessary to analyse and understand how emotions affect the maintenance of dysfunctional eating behaviours [39], using approaches and techniques that can cause behavioural changes through emotional regulation, such as positive reinforcement [39].

Generally, one of the main characteristics of patients with AN is cognitive inflexibility [75]. However, two studies included in this review that evaluated this characteristic found divergent outcomes [15,32]. Due to cognitive inflexibility, these patients may have greater difficulty adopting new therapeutic approaches, demonstrating resistance to change, even when it benefits their health [76]. Furthermore, it is essential to note that the experimental outcome devaluation approach used in this study has often been criticised for focusing on goal assessment rather than investigating habit formation [27].

In order to deepen knowledge about the mechanism underlying the learning of habits associated with food restriction in AN, recent studies have also investigated the interrelationship between the behavioural and neural mechanisms that explain such dysfunctional behaviours in these patients. By better understanding this complex relationship, we can develop more effective and personalised interventions to help those who suffer from this disorder achieve a significant improvement in their quality of life and mental health.

#### 4.1.5. Neural Mechanisms Involving Habits

Two models related to the learning process have been widely discussed in neuroscience (with regards to AN): the action–outcome learning model, which involves a neural circuit associated with reward, and the stimulus–response learning model, which includes a neural circuit related to habits [18]. Although the brain uses specific circuits for the execution of habits, other circuits are involved in carrying out actions that involve goals [77]. Despite this dichotomy, the interconnection between the two models is also possible in developing and maintaining AN.

In this sense, maladaptive eating behaviours in AN begin as rewarding actions aimed at weight loss. However, over time and with repetition, these behaviours become persistent habits that are difficult to change, playing a critical role in maintaining this disorder [18]. While pursuing thinness seems remarkably goal-oriented, the restrictive eating pattern seems immutable, even with treatment.

In the action–outcome learning model, neurotransmitters such as dopamine influence and reinforce food restriction behaviours through the sensation of pleasure, motivating the individual with AN to seek the rewarding experience again. Neuroimaging studies show increased activity in the mesolimbic circuit in response to reduced food intake, contributing to the pathological pursuit of thinness and to the reinforcement of restrictive eating behaviours [11,12,13,14,16,17,18,20,45,47], which, when repeated frequently, become less sensitive to the rewards received, no longer requiring a conscious effort to be executed.

Alterations in white matter pathways, particularly those connecting limbic structures to the cerebral cortex, may potentially be related to the psychopathology of AN [78]. However, Meneguzzo et al. [78], in their review study that focuses on a methodological issue related to the use of diffusion-weighted imaging in individuals with AN, raise the possibility of methodological bias in the data from the studies included in their meta-analysis. Therefore, the authors emphasise the importance of the careful consideration of confounding factors, with emphasis on the effect of partial volume, when interpreting neuroimaging results in patients with AN.

Stimulus–response learning is mediated by specific brain regions, such as the dorsolateral prefrontal cortex and the dorsal striatum, which play a crucial role in this process [14,20]. During this learning process, associations are formed between antecedent stimuli and subsequent responses, in which environmental stimuli trigger automatic and habitually conditioned responses [79]. However, in some instances, it can be challenging to determine whether an action was performed purely out of habit or guided by intentional, goal-directed control. Luque et al. [80] show that the results of their study are consistent with dual-process learning theories, which suggest that both goal-directed systems and habit systems operate in parallel. During each test run, specific elements of the habit system in the early stages and the goal-directed system in later stages were identified. In this sense, the final behaviour resulted from the interaction between the two learning systems. That is, response selection occurs at a later stage, where both systems are considered before a response is chosen. Therefore, both systems play an essential role in the final behaviour, and decision-making may involve a combination of the two systems.

Certain factors, such as hunger tolerance, stress, anxiety, and decreased cognitive flexibility, have a reinforcing role, and are often related to developing and maintaining food restriction in AN. These factors can influence changes in neurotransmitter systems, especially serotonin and dopamine. This may explain the difficulty that individuals with AN face in modifying their dysfunctional eating behaviours, contributing to the persistence of restrictive food choices [11,17,30,41,45,47,48]. Dopamine receives and integrates excitatory signals, which can lead to the formation of compulsive and ritualistic behaviours. According to Godier and Park [46], a repeated and prolonged increase in dopamine results in synaptic changes that may be responsible for forming compulsive habits that persist.

In this sense, compulsivity triggers repetitive and automatic behaviours that promote and maintain weight loss in AN, while anxiety can influence food choices in the search for comfort, suggesting that these individuals may find relief from anxiety around food when practising self-starvation (extreme food restriction). Restricting their food intake, they may experience temporary relief, indicating that negative reinforcement is essential in maintaining AN. Furthermore, instead of experiencing pleasure and reward when eating, dopamine in the striatum may generate discomfort, aversion, or anxiety towards food for individuals with AN. This aversive response to food may explain the persistence of food restriction observed in AN since the experience of eating is associated with aversive sensations or discomfort rather than pleasure. Thus, individuals with AN may avoid eating to avoid these negative responses [46].

Despite being in the early stages of research, some studies have investigated the use of non-invasive brain stimulation techniques, such as HF-rTMS, to modulate brain activity and modify behaviour related to food restriction in individuals with AN [28,33]. Although the precise mechanisms of the effects of treatment with HF-rTMS have not been fully clarified, Tik et al. [81] suggest that successful response to treatment in other psychiatric disorders is associated with changes in the mesocortical–limbic (dopamine circuitry) and serotonergic (mood regulation) systems. The left DLPFC plays a central role in these processes [81] and is related to the habit-learning circuit, which has been studied in individuals with AN [19,20,28,33]. Stimulation of the left DLPFC modulates ACC connectivity in a specific mesocortical limbic network [81]. Thus, HF-rTMS may promote functional brain reorganisation, altering synaptic connections and influencing habit circuits. This procedure may modulate activity in the dorsal striatum, resulting in changes in eating-related decisions in individuals with AN [28].

Understanding the mechanisms behind maladaptive food choices in individuals with AN is crucial for developing effective treatments. In this context, associative learning mechanisms play a significant role, and studies have been carried out to investigate how neural changes can affect associative learning processes, especially in the reward system [16,82].

### 4.2. Strengths and Limitations

At present, to the best of our knowledge, this is the first study to analyse research carried out to test the hypothetical habit-learning mechanisms underlying food restriction in AN. It is understood that there is a substantial need for more research that can be used to inform the development of new targets for mechanism-based therapeutic approaches. Most of the included studies presented a heterogeneous body of knowledge in the methods adopted. For example, more individuals with AN were recruited from hospitals than from outpatient clinics, and different methods of assessing habits and dietary restrictions were used, which could impair data comparability and generalisation. Furthermore, longitudinal assessments were carried out over short periods of time and may not reflect the characteristics of behaviours over time. Finally, few studies considered confounding factors and the cultural and ethnic origins of participants in their analysis. These implications highlight the need to interpret the results with caution, as they cannot be generalised without taking into account the particularities of both AN and the population being treated.

A qualitative review approach was used in this study, and a critical appraisal of the studies included was not performed. This occurred because scoping reviews aim to provide an overview of the existing literature on a given topic, without the intention of evaluating the methodological quality of the sources of evidence in these reviews [83], as scoping reviews are generally conducted to provide an overview of existing evidence, regardless of methodological quality or risk of bias [22]. Therefore, this review included articles with potential methodological biases. However, the selection of evidence sources was robust, as it included seven databases and was further enriched by the inclusion of grey literature and expert consultations.

## 5. Conclusions

This scoping review showed different methods to measure habits, including the SRHI scale, which assesses habit strength, food choice tasks, and fMRI. This methodological heterogeneity does not allow for an adequate comparability of study results. The importance of studying conditioned stimuli in the eating behaviour of patients with AN is highlighted, especially concerning the motivation for low-calorie foods and the persistent search for weight loss. These discoveries are essential for understanding the complexity of this disorder and may guide therapeutic approaches to help patients develop greater cognitive and behavioural flexibility.

Regarding neural mechanisms, the habit-learning model in AN sustains that persistent food restriction is a learned behaviour mediated by frontostriatal circuits, involving activity in the white matter, in the dorsolateral prefrontal cortex, in the basal ganglia and their structure, and in the dorsal striatum in subjects with AN compared with HCs. It is essential to consider that signals such as stress, hunger, and anxiety are associated with habit-learning.

Behavioural therapy can be a helpful approach to help individuals with AN to identify and modify these dysfunctional patterns, developing healthier and more adaptive behaviours. On the other hand, understanding the neural mechanisms behind AN is equally important, as this condition has a biological basis and is associated with changes in brain functioning. Therefore, understanding neural and behavioural mechanisms is crucial to addressing habit-based learning in individuals with AN. Thus, new directions are suggested for developing AN treatments, including behavioural interventions based on REaCH in conjunction with HF-rTMS.

Although significant progress has been achieved, some gaps still need to be investigated in future studies, such as the following: (a) How long does it take to develop a habit of food restriction and dieting? (b) Why are some people vulnerable to creating unhealthy habits related to food intake? (c) Is a certain development state (adolescence, for example) essential to developing these habits, or does it not matter if it appears in children, adolescents, or adults? Another critical issue is how to change persistent food restrictions controlled by habits. These are currently pertinent questions raised by conducting this scoping review. However, as knowledge advances and new challenges emerge, more questions will likely arise that can improve understanding and drive scientific and technological progress.

In clinical practice, this knowledge can lead to the development of evidence-based interventions aimed at modifying or breaking maladaptive eating habits. Cognitive behavioural therapies, for instance, may incorporate habit reversal techniques to disrupt harmful routines and promote healthier behaviours. Additionally, identifying specific habit-related triggers and patterns can enhance the assessment and monitoring of individuals with eating disorders, facilitating more personalised care.

For future research, investigating habits in eating disorders offers a rich avenue for exploration. Researchers can delve into the neurobiological underpinnings of habit formation and examine how they relate to the onset and progression of eating disorders. Furthermore, identifying factors that influence the development of dangerous eating habits can lead to the development of prevention strategies.

## Figures and Tables

**Figure 1 behavsci-13-00883-f001:**
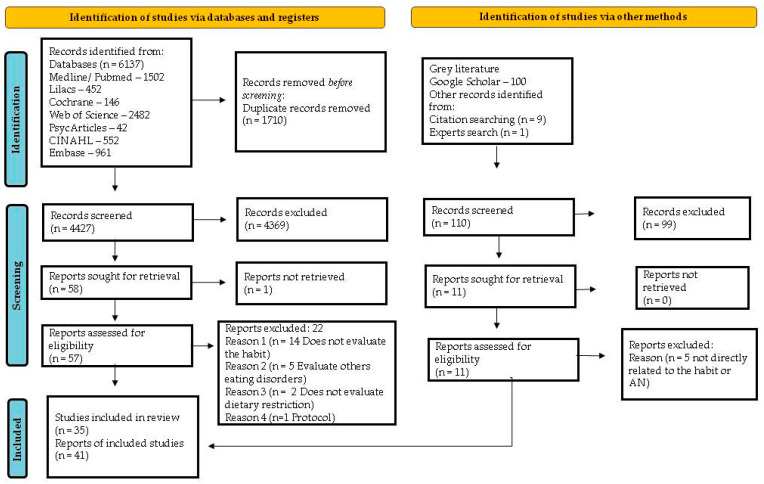
PRISMA flow diagram of included studies Page et al. [55].

**Figure 2 behavsci-13-00883-f002:**
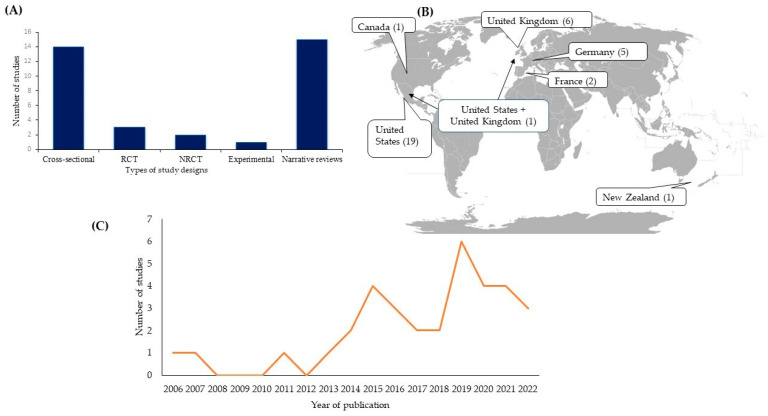
Characteristics of available evidence sources: habit-related and persistent food restriction in patients with AN. Study designs (**A**), geographic distribution (**B**), and temporal distribution (**C**).

**Figure 3 behavsci-13-00883-f003:**
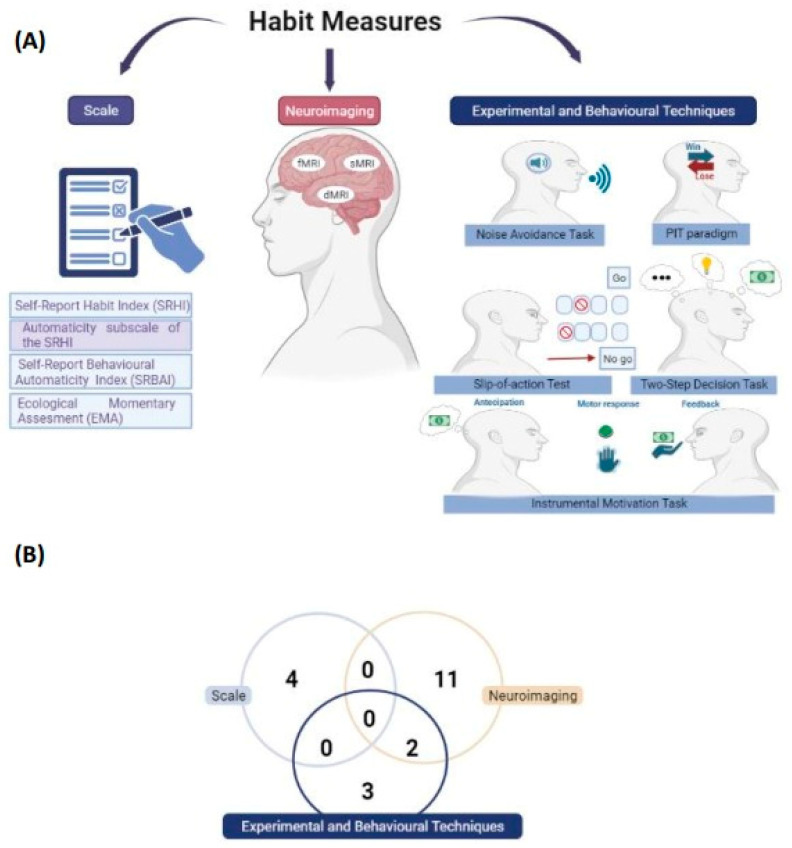
Measures to evaluate habit-related food restriction in AN. Types of methods used to assess the habit (**A**) and the number of studies according to the habit assessment measure (**B**). Created with BioRender.com, accessed on 5 September 2023.

**Figure 4 behavsci-13-00883-f004:**
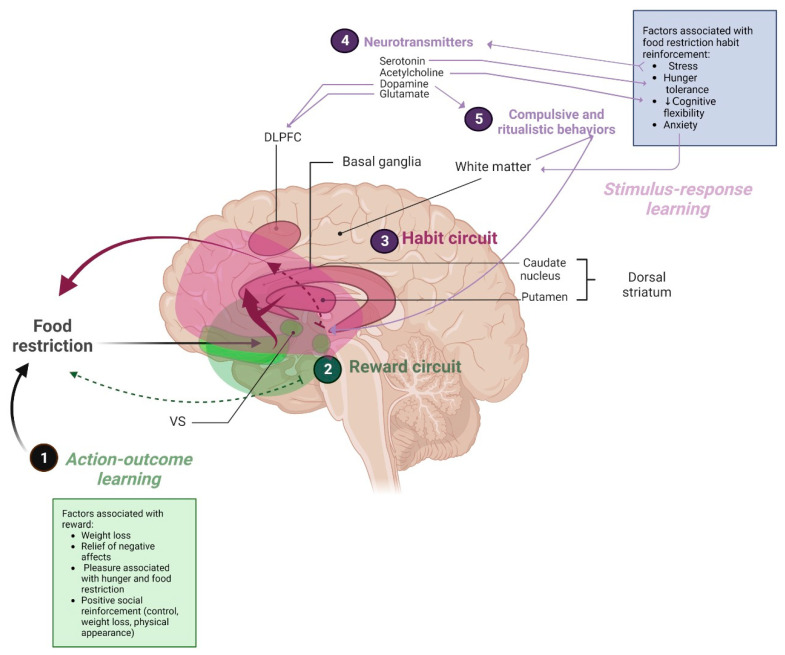
Mechanisms associated with stimulus–response learning of food restriction habits in AN. (1) Some factors, such as weight loss and alleviation of adverse effects, are elicited in the action–outcome learning circuit. (2) Food restriction engages primary reward-related circuits, including the ventral striatum (VS). (3) The stimulus–response learning circuit (habit circuit) involves the white matter, the dorsolateral prefrontal cortex (DLPFC), the basal ganglia and their structure, and the dorsal striatum (DS). (4) Factors such as tolerance to hunger, stress, and anxiety are often associated with habitual control of food restriction in AN and can lead to changes in neurotransmitters such as serotonin and dopamine. (5) Ritualistic and compulsive behaviours which contribute to sustaining habitual control via frontostriatal deactivation and abnormalities in bilateral white matter tracts. Created with BioRender.com.

**Table 1 behavsci-13-00883-t001:** Eligibility criteria for the selected studies.

Criterion	Participants	Concept	Context
Inclusion	**1. Human sample with AN:** a. Structured interviews (e.g., EDE, SCID). b. Accepted classification system (e.g., DSM, ICD). c. No age, gender, or race restrictions. **2. Animal model sample with AN.**	**Habits:** A habit is a frequently repeated behaviour elicited by a stimulus that leads to a fixed action, resulting from a learned stimulus–response association. Habit assessment in the context of AN: 1. Neuroimaging (e.g., dorsolateral striatum/putamen). 2. Scales (e.g., self-report habit index adapted for food restriction). 3. Experimental behavioural techniques (e.g., outcome/reinforcer devaluation). **Food restriction in AN:** Conceptualised as successful food avoidance or reducing caloric intake. Measures determined by authors of primary studies will be considered.	1. Primary care, hospital services, community, and remote access. 2. Any country. 3. Duration of disease and resistance to treatment. 4. Race, gender, and socioeconomic status.
Exclusion	1. Risk behaviours for eating disorders. 2. Overweight/obesity 3. Other eating disorders (e.g., bulimia nervosa, binge eating disorder). 4. Pregnant women and nursing mothers.	**Habits:** 1. Outside the context of AN. 2. Usual physical activity. 3. Habitual meal and food intake with no focus on food restriction. **Food restriction:** 1. Outside the context of AN. 2. Professional-oriented diet for specific diseases (e.g., obesity, diabetes). 3. Dietary intakes in AN assessed by daily energy/nutrients or dietary patterns.	None.

AN—anorexia nervosa; EDE—eating disorder examination; SCID—structured clinical interview for the DSM-IV; DSM—*Diagnostic and Statistical Manual for Mental Disorders*; ICD—International Statistical Classification of Diseases and Related Health Problems.

## Data Availability

Data are contained within the article, Appendix A, and protocol https://osf.io/d2jns (accessed on 8 January 2022).

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
