# Peer review of "Habits and Persistent Food Restriction in Patients with Anorexia Nervosa: A Scoping Review"

_behavsci, 2023, doi:10.3390/bs13110883_

Round 1

Reviewer 1 Report

This scoping review on "Habits and Persistent Food Restriction in Patients with Anorexia Nervosa" provides a comprehensive overview of the current state of research in this critical area of eating disorders. The study is commendable for its effort to compile and analyze existing literature on the topic, shedding light on the complex nature of anorexia nervosa and its enduring impact on individuals.

One notable strength of this review is its thorough examination of various habits and behaviors associated with anorexia nervosa, including patterns of food restriction. The authors have skillfully synthesized findings from diverse sources, offering valuable insights into the multifaceted aspects of this eating disorder. By presenting a wide range of data, the study underscores the need for a nuanced understanding of the condition.

However, it is crucial to acknowledge certain limitations in this scoping review. First and foremost, the authors should have highlighted the heterogeneity and potential biases within the included studies. Anorexia nervosa is a highly individualized disorder, and the variation in study populations and methodologies can significantly affect the generalizability of findings. Furthermore, the review lacks a critical appraisal of the quality of the studies analyzed, which is vital for assessing the reliability of the evidence presented.

Additionally, the review could benefit from a more focused discussion on the implications of these findings for clinical practice and future research directions. While it offers a comprehensive synthesis of existing knowledge, it falls short of providing concrete recommendations or highlighting gaps in the literature that warrant further investigation.

In conclusion, "Habits and Persistent Food Restriction in Patients with Anorexia Nervosa: A Scoping Review" is a valuable resource for researchers and professionals seeking an overview of the current state of research in this area. However, it is imperative for readers to consider the limitations and potential biases inherent in the included studies. The review serves as a useful starting point for future research endeavors aimed at enhancing our understanding of anorexia nervosa and its treatment.

Author Response

Consulte o anexo

Reviewer 2 Report

The manuscript reports a review of the literature about habits and persistent food restriction in anorexia nervosa patients. The paper is well written and covers an interesting area of research. The methods followed the international guidelines and are well described. I do not have specific concerns. I just have a few comments that could improve the manuscript:

- please report  the key search in the text

- have you evaluated the quality of the studies included in your review?

- is it possible that alteration in limbic and sensorimotor systems found in the literature have a role in the development of habits (see https://doi.org/10.1002/eat.23160)?

- Despite habits, what is the role of socio-cultural roles in your model?
